# Enhanced carbon dioxide conversion at ambient conditions via a pore enrichment effect

Wei Zhou[1,5], Qi-Wen Deng[1,2,5], Guo-Qing Ren [1], Lei Sun [1✉], Li Yang [1,3,4], Yi-Meng Li[1], Dong Zhai [1], Yi-Hong Zhou[2] & Wei-Qiao Deng [1,3✉]

Chemical fixation of carbon dioxide ($CO_2$) may be a pathway to retard the current trend of rapid global warming. However, the current economic cost of chemical fixation remains high because the chemical fixation of $CO_2$ usually requires high temperature or high pressure. The rational design of an efficient catalyst that works at ambient conditions might substantially reduce the economic cost of fixation. Here, we report the rational design of covalent organic frameworks (COFs) as efficient $CO_2$ fixation catalysts under ambient conditions based on the finding of "pore enrichment", which is concluded by a detailed investigation of the 10994 COFs. The best predicted COF, Zn-Salen-COF-SDU113, is synthesized, and its efficient catalytic performance for $CO_2$ cycloaddition to terminal epoxide is confirmed with a yield of 98.2% and turnover number (TON) of 3068.9 under ambient conditions, which is comparable to the reported leading catalysts. Moreover, this COF achieves the cycloaddition of $CO_2$ to 2,3-epoxybutane under ambient conditions among all porous materials. This work provides a strategy for designing porous catalysts in the economic fixation of carbon dioxide.

[1] Institute of Molecular Sciences and Engineering, Institute of Frontier and Interdisciplinary Science, Shandong University, Qingdao 266237, P. R. China. [2] Collage of Hydraulic & Environmental Engineering, China Three Gorges University, Yichang, Hubei 443002, P. R. China. [3] State Key Laboratory of Molecular Reaction Dynamics, Dalian National Laboratory for Clean Energy, Dalian Institute of Chemical Physics, Chinese Academy of Sciences, Dalian 116023, P. R. China. [4] University of the Chinese Academy of Sciences, Beijing 100039, P. R. China. [5] These authors contributed equally: Wei Zhou, Qi-Wen Deng. ✉email: slei@sdu.edu.cn; dengwq@sdu.edu.cn

As an important component of the air, $CO_2$ is well known as a greenhouse gas. With the continuous consumption of fossil fuels, the $CO_2$ concentration in the atmosphere reached 410 ppm in 2017[1], which was the highest value in human history, and the concentration continues to increase. The resulting greenhouse effect has started to affect the stability of global ecosystems, leading to the frequent occurrence of global extreme weather. Currently, there are two industrialized technologies to reduce $CO_2$ emissions, capture or conversion, which already have been working at annual ten-million-ton scale. However, either capture or conversion technology imposes a high cost because of the facilities requiring the high temperature or pressure or both. The global economy cannot afford billion-ton scale $CO_2$ emission reduction. Therefore, a method to capture and convert $CO_2$ at ambient conditions is a potential solution for economic $CO_2$ emission reduction.

In 2013, our previous work found a Salen-conjugated microporous polymer (CMP) can capture and convert $CO_2$ into valuable chemicals at ambient conditions[2]. As far as we know, metal-Salen catalysts can realize the $CO_2$ transformation under relatively mild conditions[3–7]. Moreover, the pore structure in CMP can further lower the requirements of reaction temperature and pressure. With the goal to improve the reaction conditions, extensive efforts have been dedicated by combining different moieties possessing high catalytic activity and the porous polymer and framework materials, such as porous organic polymers (POPs)[8], microporous organic polymer (MOP)[9] and metal-organic frameworks (MOFs)[10–14]. Besides, the metal-free ionic polymers have also been applied in the syntheses of porous catalytic materials under the mild conditions[15,16]. All these researches indicate that the incorporation of catalytic active moieties into the porous materials is a feasible way to realize the fixation $CO_2$ under ambient conditions by achieving capture and conversion of $CO_2$ simultaneously. The CMPs, POPs, MOPs, MOFs, etc. are actually library compounds with millions of possible candidates. There is no doubt that a better candidate can be found by the large-scaled screening method, which has been used in various purposes in literatures[17–25].

In this work, we use large-scale computation screening method to address the best candidate in a constructed library of COFs and demonstrate the best predicted COF. First, we build a virtual library, including 10994 COF structures using Salen-metal (M-Salen) as the catalytic centre. The $CO_2$ adsorption performances at 298 K and 1 bar are theoretically investigated. Based on the simulation results, we analyse the $CO_2$ distribution characteristic within the frameworks of the materials with the best and worst $CO_2$ adsorption performance. Second, we find a pore enrichment effect by observing the enriched local concentration of $CO_2$ inside special pore structures. With this pore enrichment effect, the reaction rate can be significantly increased, leading to the efficient catalytic process. Finally, we synthesize the best predicted COF structure (abbreviated as Zn-Salen-COF-SDU113) and measure its catalytic performance for $CO_2$ molecule fixation. The obtained COF exhibits excellent $CO_2$ conversion performance under atmospheric pressure and room temperature. The Zn-Salen-COF-SDU113 catalyst shows high catalytic activity and excellent stability and recyclability towards terminal epoxides (PO) and internal epoxides (2,3-epoxybutane) under ambient conditions.

## Results

**Simulation prediction**. To construct a structure library of Salen-COFs, we choose three types of linker groups to build different topological structures, as shown in Fig. 1. Then, twenty-seven metal elements are used as the metal centre of M-Salen, and the alkane substituent groups usually present in the Salen

molecule are also considered in the construction process of the Salen-COFs. Finally, 10994 M-Salen-COF structures were built for the structure dataset. The excess adsorption amounts of $CO_2$ by the 10994 M-Salen-COF structures, which were optimized by molecular dynamic methods, were simulated. Most of the excess value is distributed in the range from 0.5 to 1.5 mol $g^{-1}$ and the excess amounts for 252 structures are higher than those of Co-CMP (1.802 mol $g^{-1}$), which can capture and convert $CO_2$ under ambient conditions[2]. For all COFs, the maximum value of 3.99 mol $g^{-1}$ is for Zn-Salen-COF-SDU113. It is worth noting that the second-highest value (3.94 mol $g^{-1}$) belongs to the No.117 COF structure with the Fe-Salen compound. Moreover, the other six highest excess amounts are all for COFs with the No. 117 COF structure. The metal elements in the M-Salen compound are Co, Cr, Mo, Ru, Re and Au, which show excess amounts of 3.83, 3.82, 3.75, 3.72, 3.58 and 3.47 mol $g^{-1}$, respectively. According to the analyses of the $CO_2$ average density shown in Fig. 2, the captured $CO_2$ molecules in these seven COFs with the same COF structure are mainly concentrated between the two linker groups along the direction of the 2D pore. This type of linker group, which is a large conjugated structure, provides the capture environment for $CO_2$. The result of adsorption between two layers is that the introduction of the corresponding metal atom promotes the structural deformation of the No.117 COF. Although the structure leads to $CO_2$ enrichment, only a small portion of the captured $CO_2$ molecules can directly interact with the M-Salen compound in these COFs. Similarly, $Fe_{108}$ exhibits the same distribution feature. In the constant $Zn_{113}$ (Zn-Salen-COF-SDU113) and $Zn_{370}$, some of the captured $CO_2$ molecules gather around the Zn-Salen fragment, which is the catalytic reaction centre in the chemical fixation of $CO_2$. For these M-Salen-COFs with high $CO_2$ uptakes, we found a pore enrichment effect. Precisely due to the enrichment of $CO_2$ molecules, the excess adsorption amount is improved significantly. This gathered phenomenon in the pore may change the reaction environment of the chemical fixation of $CO_2$.

To further state the pore enrichment effect, we also analysed the average densities of $CO_2$ for the COFs with the worst adsorption performance, as shown in Fig. 2. The ten COFs exhibit the low excess adsorption amounts of 0.277, 0.297, 0.306, 0.309, 0.312, 0.315, 0.328, 0.330, 0.334 and 0.339 mol $g^{-1}$ for $Hg_{148}$, $Tl_{107}$, $Os_{117}$, $Pt_{117}$, $Bi_{107}$, $Pb_{117}$, $Sb_{107}$, $Ir_{311}$, $Cd_{117}$ and $Re_{117}$/ Salen-COFs, respectively. The results of the $CO_2$ average densities clearly indicate that the captured $CO_2$ molecules are evenly distributed around the COFs. In these COFs, there is no strong enough molecular interaction to effectively capture the $CO_2$ molecules. Interestingly, three Salen-COFs ($Tl_{107}$, $Bi_{107}$ and $Sb_{107}$) with the No. 117 COF are among the ten worst COFs. In these three COFs, only a few $CO_2$ molecules were distributed between the linker groups along the direction of the 2D pore, and $CO_2$ enrichment is very weak. The measurement of the distance between the two linker layers for $Au_{107}$ and $Tl_{107}$ shows that the interlayer spacing decreases from 7.29 to 5.36 Å, thus blocking $CO_2$ enrichment between the two adjacent conjugated groups. The above analysis on the ten M-Salen-COF structures with the highest excess amount shows that a "pore enrichment effect" obviously enhances the $CO_2$ adsorption performance. But, no enrichment is not found in the low $CO_2$ uptake structures. The difference indicates that the pore enrichment is benefit to the $CO_2$ storage in the porous materials.

Besides the influence on the uptake amount, we further study the pore enrichment effect on the catalytic reaction. The external and internal pressures of the gas-filled $Zn_{113}$ and $Zn_{321}$ were compared. In $Zn_{113}$, there is obvious $CO_2$ enrichment, but the captured $CO_2$ in $Zn_{321}$ is evenly distributed around the framework. Figure 3b shows that the internal pressure increases with

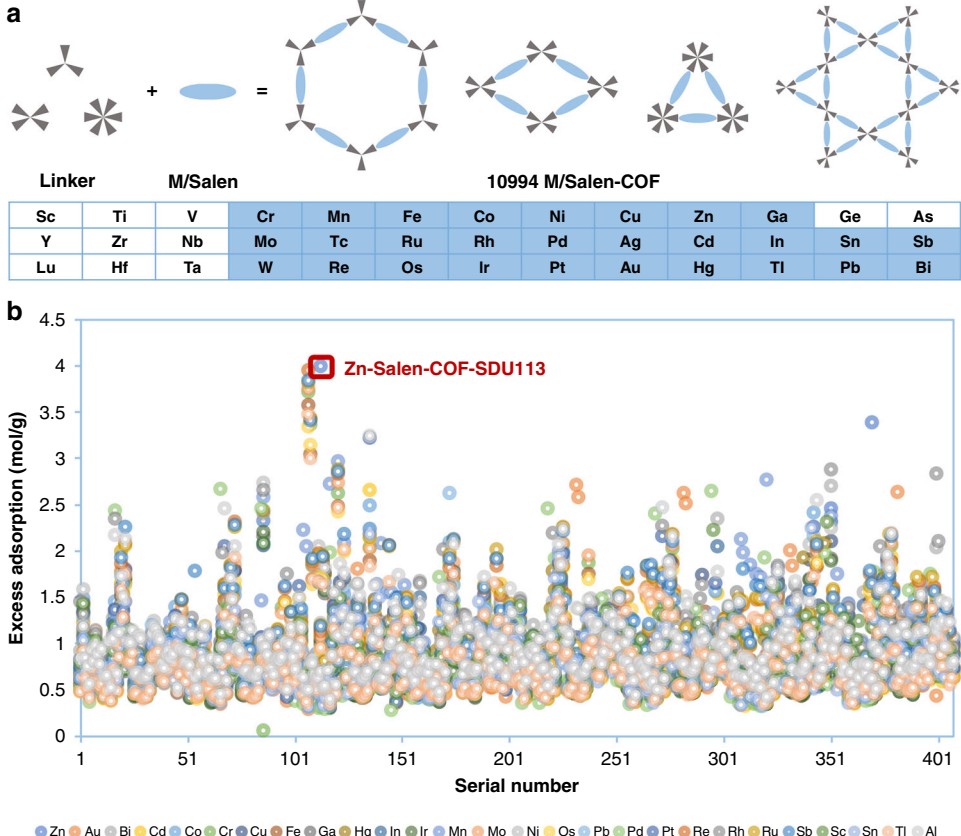

| Sc | Ti | V | Cr | Mn | Fe | Co | Ni | Cu | Zn | Ga | Ge | As |
|----|----|----|----|----|----|----|----|----|----|----|----|----|
| Y | Zr | Nb | Mo | Tc | Ru | Rh | Pd | Ag | Cd | In | Sn | Sb |
| Lu | Hf | Ta | W | Re | Os | Ir | Pt | Au | Hg | Tl | Pb | Bi |

**Fig. 1 Topology diagram and excess CO₂ adsorption amount. a** The polygon skeletons and the metal element types for the 10994 kinds of Metal/Salen-COFs. Three types of linker groups and Metal/Salen build four topology structure, which from left to right are hexagonal, single-pore rhombic, trigonal and dual-pore kagome, respectively. The chemistry structures for all the linker groups were listed in the Supplementary Information. Twenty-eight elements were chosen as the central metal of metal/salen. **b** The simulated excess adsorption amount (mol g$^{-1}$) of CO$_2$ in 10994 kinds of Metal/Salen-COFs at 298 K and under 1 bar.

the external pressure. When the external pressure is 1 bar, the internal pressure for $Zn_{113}$ and $Zn_{321}$ is 25.3 and 4.97 bar, respectively. As the external pressure increases to 10 bar, the internal pressure reaches 57.2 and 36.4 bar, respectively. The gap between $Zn_{113}$ and $Zn_{321}$ indicates that the pore enrichment effect favours an increase in the gas pressure in the pore. We further studied the pore enrichment effect on the reaction rate ($k_{PC}$) of the coupling reaction of CO$_2$ and propylene oxide (PO) into propylene carbonate (PC), based on the rate Eq. (1) as follow,

$$k_{PC} = A exp\left(\frac{-E_a}{RT}\right)[CO_2]^m[PO]^n \tag{1}$$

where, $A$ is the pre-exponential factor, $E_a$ is the activation energy, $R$ is the gas constant, $T$ is the temperature, [CO$_2$] and [PO] is the concentration of reactants, and $m$ and $n$ is the reaction order. Here, they both equal to 1. Additionally, because the central metal elements of the catalysts are both zinc and the species of reactants are the same, we assumed that $E_a$ and $A$ in the catalytic processes using $Zn_{113}$ and $Zn_{321}$ are the same, thus the reaction rate can be approximately only proportional to [CO$_2$] inside the pore. We compared the increment values ($Ik$) of $k_{Zn113}$ and $k_{Zn321}$ under internal pressure to the ones under external pressure, respectively, as shown in Fig. 3c, d. As a whole, when the external pressure increases gradually, the pore enrichment effect will weaken. However, at 1 bar external pressure, $Ik_{Zn113}$ is the highest (29.3 times) and $Ik_{Zn321}$ is only 5.07. As the external pressure increases, $Ik_{Zn113}$ and $Ik_{Zn321}$ fall to 9.13 and 4.39, respectively. This result shows that the increment change of $k_{Zn113}$ is more obvious, especially at low pressure. But for $Zn_{321}$,

the increment changes a little from 1 to 10 bar. It reflects that the pore enrichment effect in $Zn_{113}$ is much stronger than $Zn_{321}$ as shown in the analysis results of CO$_2$ average densities. Due to the obvious pore enrichment effect, it is easy for $Zn_{113}$ to implement catalytic reactions requiring high-pressure conditions under atmospheric pressure.

It is worth nothing that the pore enrichment effect is different from the confinement effect which mainly acts on the electronic structure of the reactants or products, thus changes the energy barrier of the catalytic processes. And yet, the pore enrichment effect mostly caused by the behaviour of numerous molecules in the pore. We will expound the differences in the later 'Discussion' section.

**Experiment verification.** Based on the simulated results above, the corresponding experiments have been carried out to verify the pore enrichment effect. We synthesized the best Salen-COF, Zn-Salen-COF-SDU113, by using the similar synthesis method[2,26–28]. Both Fourier transform infrared (FT-IR, Supplementary Figs. 1 and 2) and soild state $^{13}$C nuclear magnetic resonance ($^{13}$C NMR, Supplementary Fig. 3) measurement results indicate the formation of imine bonds via Schiff condensation of the ligand TTHEPB and EDA[20,21]. The experimental Powder X-ray diffraction (PXRD, Supplementary Fig. S4) pattern matched well with the simulated pattern constructed from the AA stacking of the 2D layers (Fig. 4b). The unit cell parameters were also obtained ($a = b = 43.887$ Å, $c = 8.418$ Å, $\alpha = \beta = 90°$, $\gamma = 120°$, $R_p = 6.98\%$ and $R_{wp} = 4.95\%$, for

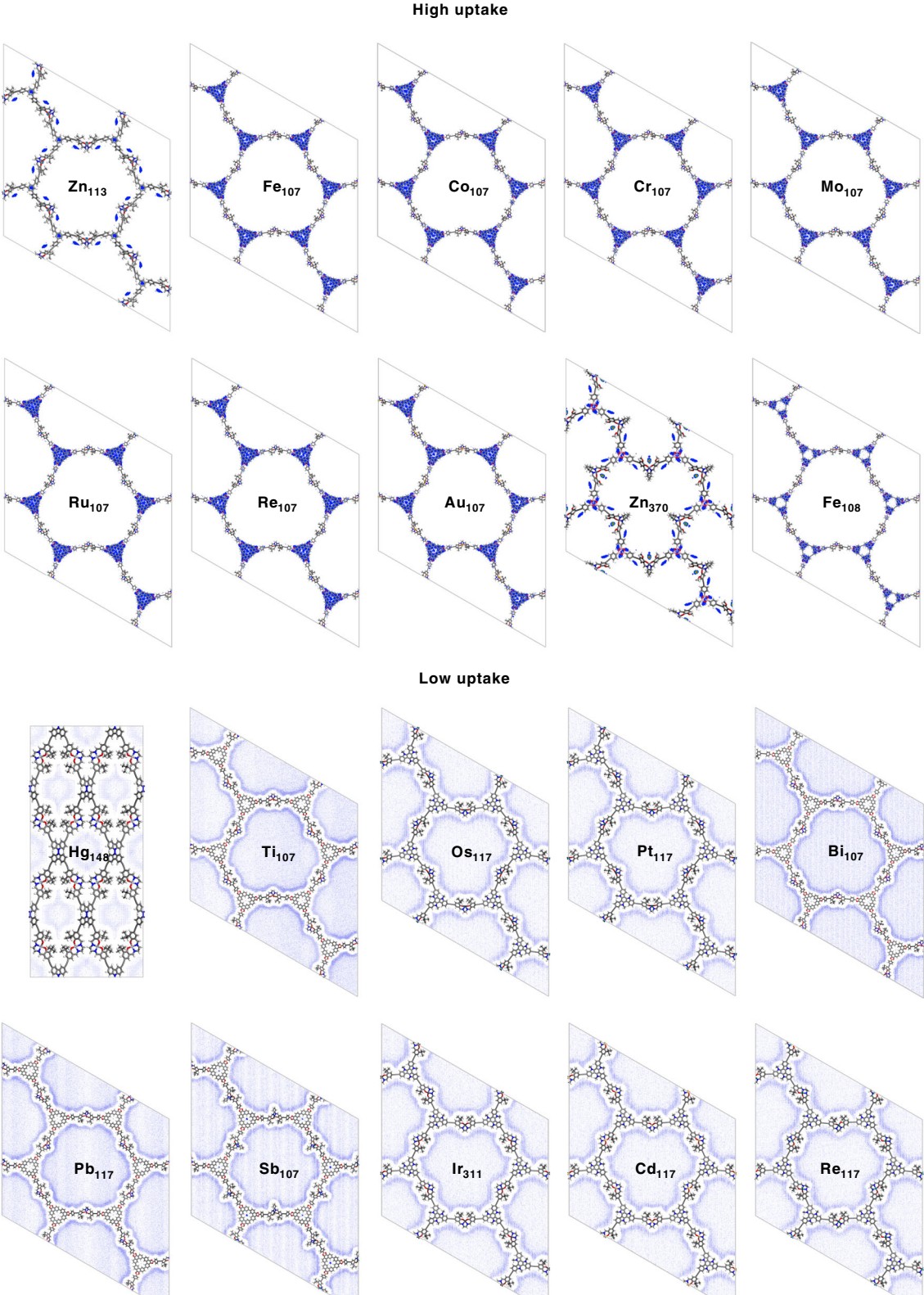

**Fig. 2 CO$_2$ average densities.** Ten Metal/Salen-COFs with the highest excess CO$_2$ adsorption amount (upper) and ten Metal/Salen-COFs with the lowest excess CO$_2$ adsorption amount (lower). The different shades of colour distinguish the degree of CO$_2$ uptake. Dark blue represents a large average density.

more details, see Supplementary Table 2). As shown in Supplementary Fig. 5, scanning electron microscopy (SEM) images revealed that all Zn-Salen-COF-SDU113 powders retained a spherical morphology with an average particle size of 0.5–2 μm.

It is worth noting that the transmission electron microscopy (TEM) images exhibit a regular channel structure (Fig. 4c); those images exhibit a uniform honeycomb structure, which is in good agreement with the predicted structure. The measured average

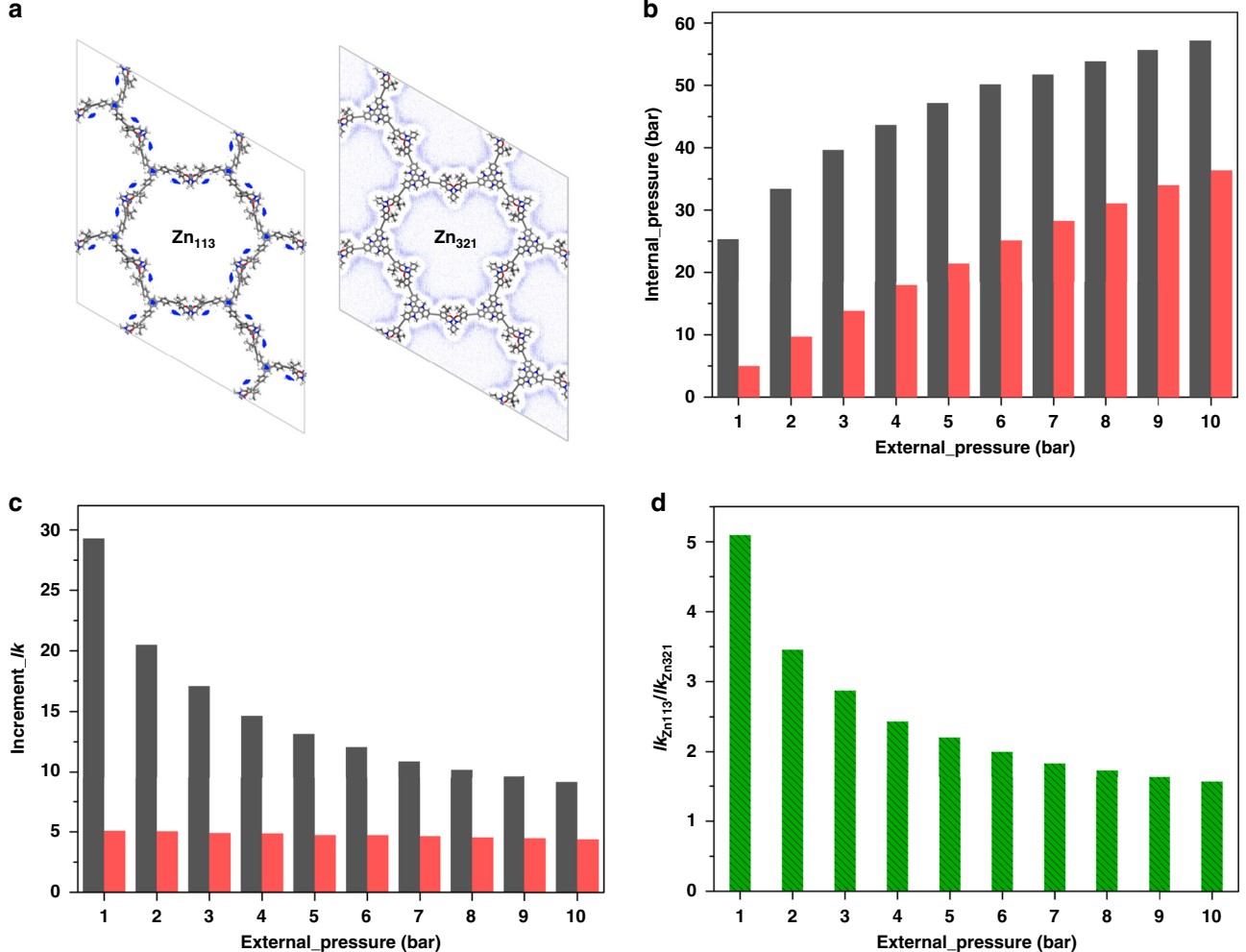

**Fig. 3 Pore enrichment effect on the internal pressure and reaction rate. a** $CO_2$ average densities of in Zn-Salen-COF-SDU113 and 321. **b** The variation trend of internal pressure along with increasing external pressure. **c** The increment of the reaction rate ($Ik$) for Zn-Salen-COF-SDU113 ($Ik_{Zn113}$) and 321 ($Ik_{Zn321}$) under internal and external pressure, respectively. **d** The ratio of $Ik_{Zn113}/Ik_{Zn321}$. In **b**, **c**, the black and red colours represent Zn-Salen-COF-SDU113 and 321, respectively.

diameter of those channels is ~3.5 nm, which also matches the distance between two adjacent pore walls (incident electron beam perpendicular to the channels, Fig. 4c). The pore structure was explored by nitrogen adsorption and desorption isotherms at 77 K and $CO_2$ adsorption analyses at 298 K (Supplementary Figs. 7 and 8). The nitrogen adsorption isotherms revealed uptake below $P/P_0 = 0.05$. The Zn-Salen-COF-SDU113 samples have a BET surface area of 1143 $m^2\,g^{-1}$. When $P/P_0$ approaches 0.0167, the $CO_2$ uptake reaches 19.8 $mg\,g^{-1}$. According to the nonlocal density functional theory, the pore size distribution of Zn-Salen-COF-SDU113 is 3.5 nm, indicating that it is a mesoporous material. Moreover, the pore size distribution is in good agreement with the pore diameter optimized space-filling model and the average diameter measured from the HRTEM images also match the simulative pore diameter and pore size distribution. We notice that part of the channel structure is distorted, this fact suggests structural changes that are probably caused by the activation process and high vacuum and high energy electron beam conditions during HRTEM testing[29]. Based on the above results, Zn-Salen-COF-SDU113 with **hcb** topology was successfully synthesized. By increasing the dosage of organic ligands by 20 times, we found that the synthesized Zn-Salen-COF-SDU113 can still keep good crystallinity. As shown in Fig. 4e–g, the EDS mapping proved the homogeneous

distribution of $Zn^{2+}$ ions in the crystal structures. Moreover, the ICP analysis shows that the Zn element content is 4.18 wt%.

Zn-Salen-COF-SDU113 demonstrates excellent activity and selectivity for catalysing the reaction of $CO_2$ and terminal epoxides (PO) into PC under ambient conditions. After metalation with $Zn^{2+}$, the obtained Zn-Salen and Zn-Salen-COF-SDU113 powders were mixed with the $nBu_4NBr$ co-catalyst (TBAB), and then, the reaction system was purged three times with a $CO_2$ atmosphere. This reaction was performed in pressure tubing at 1 bar and 298 K. As shown in Supplementary Table 3, Entries 1–5 show the catalytic performances of different catalysts under the same experimental conditions. The test results indicate that the co-catalyst TBAB plays a key role in catalytic conversion of $CO_2$ and PO. The coupling reaction between $CO_2$ and PO catalysed by Zn-Salen-COF-SDU113 yields 10.3% PC without the co-catalyst TBAB. However, the co-catalyst TBAB showed a yield of 20.4% in our previous work[1]. When mixed with TBAB, Zn-Salen-COF-SDU113 can effectively accelerate the catalytic reaction with an improved yield of 98.2% and a turnover number (TON) value of 3068.9, which are much higher than those of Zn-Salen tested under the same conditions (Yield: 79.2%, TON: 162.3). Fig. 5a shows the relationships between the yield and TON (PC) and reaction time at atmospheric pressure and room temperature. It is obvious that both the yield and TON values

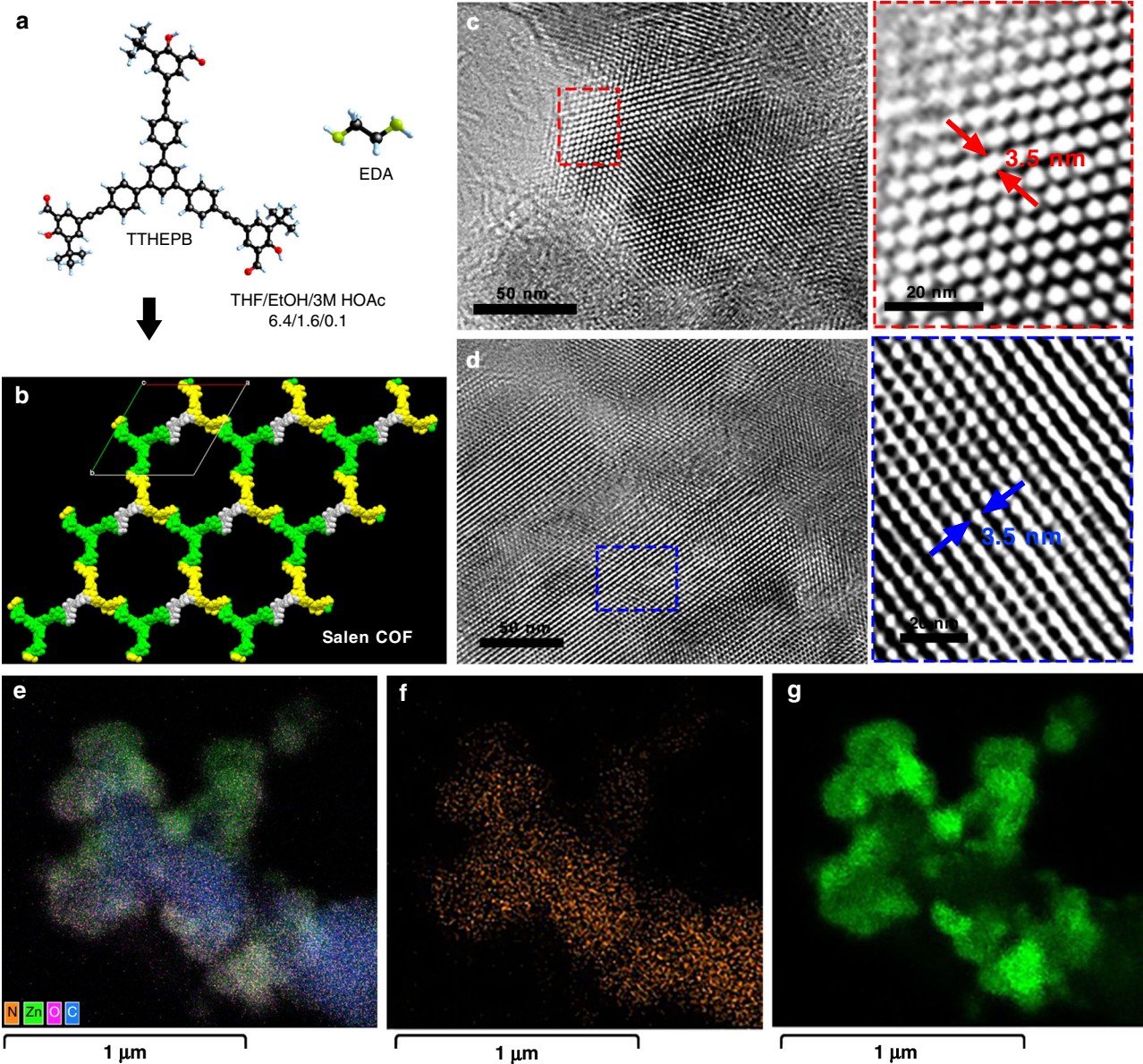

**Fig. 4 Synthesis and structure of Zn-Salen-COF-SDU113. a, b** Synthesis of Zn-Salen-COF-SDU113. **c, d** TEM images of Zn-Salen-COF-SDU113 (the dotted border of different colours corresponds to different areas). **e–g** Energy-dispersive X-ray spectroscopy (EDS) mapping images of Zn-Salen-COF-SDU113 (orange: N elements; green: Zn elements).

are proportional to the reaction time. To further confirm that our simulation studies are appropriate and useful in guiding the COF design, another COF (Zn-Salen-COF-SDU11) with lower expected performance to Zn-Salen-COF-SDU113 were synthesized for comparison. As is consistent with simulation studies, under the same reaction conditions, Zn-Salen-COF-SDU11 shows catalytic activity with a yield of 84.2% and a turnover number (TON) value of 1142.7, which are much lower than Zn-Salen-COF-SDU113 (Yield: 98.2%, TON: 3068.9). The trend of theoretical calculation is consistent with that of actual catalytic measurement. Moreover, upon further exploring the recycling stability of Zn-Salen-COF-SDU113, it is obvious that there is no significant decrease in the yield after 5 cycles of the $CO_2$/ PO coupling reaction. Moreover, the obtained Zn-Salen-COF-SDU113 still maintains good crystallinity after five cycles (for more details, see the PXRD patterns in Supplementary Fig. 4), indicating the excellent stability and recyclability of the solid COF catalyst during the catalytic $CO_2$/PO reaction. Moreover,

we also test the catalytic activity towards internal epoxides (2,3-epoxybutane) with a substantial yield of 74.4% and a TON value of 533.9 at ambient conditions, which have never been achieved under ambient conditions by other porous materials including CMPs, POPs, MOPs and MOFs.

## Discussion

For our designed M-Salen-COFs, the embedded catalytic moieties retain the catalytic activity for $CO_2$, and the porosity of the framework provides an environment space for gas capture. Variety of viable porous materials can be used based on this concept[29–33]. It is worth noting that the embedded strategy is not only the superposition of two functions. The chemical reactions are also affected by the pore environment in the porous materials, such as the confinement effect. The confinement effect indicates that the electronic orbitals and reaction barrier of the pore structure will change when the molecules enter into the

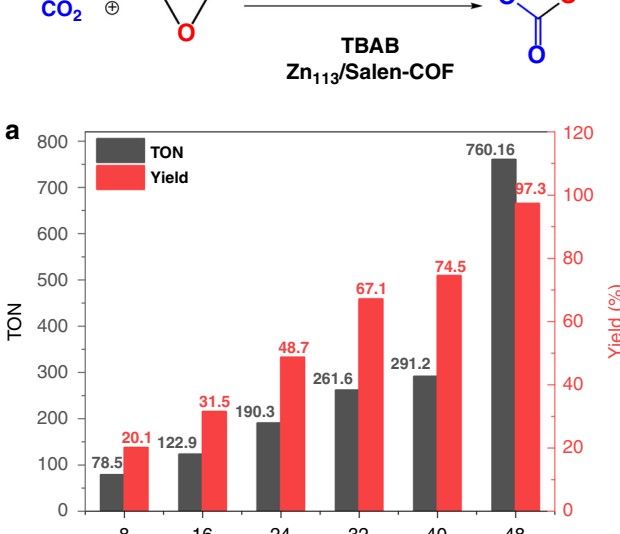

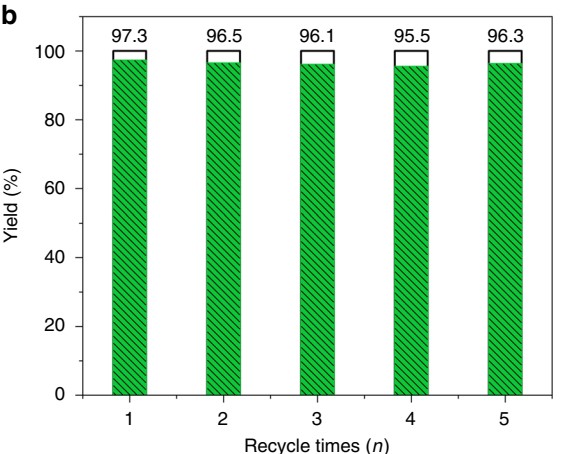

**Fig. 5 Catalytic performance for Zn-Salen-COF-SDU113. a** The yield of PC depending on the reaction time and **b** recycling stability of Zn-Salen-COF-SDU113 at atmospheric pressure and room temperature.

pores of zeolite, affecting chemical reactions[34–36]. It was found that the confinement effect can be synergistically cooperate with nano effect to catalysis reactions[27,37–39]. The loaded nanoparticles inside carbon nanotubes change the catalytic performance of the nanoparticles. The analysis results show that the relatively narrow pores significantly increased the catalytic activity. In recent years, people have extended this concept. It is believed that the reactions in porous materials with proper pore size, including zeolites, MOFs, COFs and other molecular cage materials, can be all affected by the pore environment and change the electronic orbital or charge distribution of the reactive molecules to change the reaction barrier[40–42]. In general, the electronic confinement more likely to happen when the size of the adsorbate molecule is similar to the size of the pore[34]. With the narrow or crowded pores relative to the molecule size, the molecular structures of the adsorbate are more easily affected by the confined environment, so the confinement effect is the dominant factor. When the pore size increases or the adsorbate size decreases, we believe that the confinement effect should gradually weaken because the adsorption space is large enough

for the adsorbate under the circumstances[43], and the pore enrichment effect caused by the molecular aggregation turn into the most important factor. For the $CO_2$ adsorption in the M-Salen-COFs designed by us, the enrichment just changes the concentration of reactants around the catalytic centre, not the obviously electronic structure of reactant. It is an effect of changing the equilibrium conditions of the catalytic reaction. Therefore, the pore enrichment will promote the thermodynamic process for the small-size molecule catalysed by using relatively large-pore porous materials. We infer that the reactions involving with gases is the most common chemical reaction. Therefore, we think that the pore enrichment effect can be widely applied for the small-size gas molecules, hydrogen, oxygen and ethylene so on.

In summary, we built about ten thousand of M-Salen-COF structures using Salen-metals as the catalytic centre, and their corresponding $CO_2$ capture performances were theoretically investigated by studying the $CO_2$ distribution, $CO_2$ adsorption abilities and molecular aggregation. According to the simulated results, we synthesized the corresponding COF materials (Zn-Salen-COF-SDU11, Zn-Salen-COF-SDU113 and Zn-Salen-COF-SDU128), and it exhibited outstanding catalytic PO/$CO_2$ coupling reaction activity and excellent stability and recyclability. Moreover, Zn-Salen-COF-SDU113 can even transform internal epoxides (2,3-epoxybutane) under ambient conditions. Based on the theoretical and experimental results, we find the pore enrichment effect to explain the effect on the $CO_2$ fixation process caused by the molecular aggregation state. This work provides an effective strategy, by using pore enrichment effect, to achieve excellent catalysts with high catalytic activity, chemical stability and recycling ability by combining simulations and experiments. This strategy provides a direction for the oriented design of COFs and other porous materials for $CO_2$ economic fixation.

## Methods

**Simulation methods**. To obtain stable frameworks, we optimize the geometries for each COF through annealing method. The classical molecular dynamics package LAMMPS[43] is used with the universal force field (UFF)[44]. A periodic unit cell is employed. In simulated annealing, the structures are first heated to and equilibrated at 500 K for 1 ns. They are then cooled to 300 K at a rate of 25 K/100 ps. The last step is equilibration at 300 K for 1 ns. Based on the dynamic trajectories of equilibration, we select the lowest-energy geometry as the starting geometry in the next annealing process. After repeated cycling, the optimization is completed when the energy convergence threshold reaches $1 \times 10^{-4}$ kcal/mol. The NVT ensemble is used during annealing, with a Nose–Hoover thermostat for temperature and volume control. To determine the $CO_2$ adsorption capacity, the grand canonical Monte Carlo (GCMC) method was used with Dreiding force field. To obtain an accurate measure of molecule loading, we specify 1,000,000 equilibration steps before the production stages and 3,000,000 Monte Carlo steps in the production stage of the simulation. Charges in the GCMC simulations are set through the QEq charge equilibration method. Excess uptake is defined as the number of $CO_2$ molecules that can be adsorbed around the pore wall of porous materials compared to the free volume[45,46].

**Synthesis and characterization methods**. The Salen-COF powder were synthesized by traditional solvothermal method. In typically, 50 mg TTHEPB were dissolved in a mixture of 6.4 ml THF and 1.6 ml EtOH. Then, 30 μL EDA was added, the mixed solution quickly turned green, 0.1 ml HOAc (3 M) is immediately added to the above solution. The solution was further moved into a 10 mL pressure tube, after three freeze-pump-thaw cycles, the pressure tube was heated at 120 °C for 3 days. The solid was collected by centrifugation and washed with dried THF and acetone. The yellow powder was then dried at 80 °C under vacuum for 24 h to give the Salen-COF in 51% isolated yield (76 mg, based on the total weight of TTHEPB and EDA). Anal. Cald for $(C_{22}H_{19}NO)n$: C 80.66; H 6.71; N 5.89. Found: C 79.03; H 6.39; N 5.24. To expand the scale of Salen-COF synthesis, the dosage of TTHEPB was broadened to 0.3 mmol. The obtained powder also exhibits high crystallinity (Supplementary Fig. 5). To a mixture of Salen-COF (200 mg) and Zn (OAc)$_2$ (50 mg), was added dry MeOH (25 mL) under N$_2$ and stirred at room temperature for 48 h. The resulting precipitate was further obtained by filtration and washed with THF, acetone to afford Zn-Salen-COF-SDU113. Moreover, we have synthesized the identified COF (Zn-Salen-COF-SDU113) on a larger scale and addressed an approach to enlarge the production of Zn-Salen-COF-SDU113. For the synthesis of

1.02 g COF, 1 g TTHEPB were dissolved in a mixture of 32 ml 1,4-dioxane and 8 ml EtOH. Then, 200 μL EDA was added, the mixed solution quickly turned green, 2 ml HOAc (3 M) is immediately added to the above solution. The solution was further moved into a 100 mL stainless autoclave and heated at 120 °C for 3 days. The solid was collected by centrifugation and Soxhlet extraction with dried 1,4-dioxane and acetone. The powder was then dried at 80 °C under vacuum for 24 h to give the Salen-COF in 85% isolated yield. The synthetic details for Zn-Salen-COF-SDU11 is similar to Zn-Salen-COF-SDU113. (see Supplementary Table 1).

**Catalytic activities evaluation.** The Zn-Salen or Zn-Salen-COF-SDU113 catalysts were dried at 120 °C in vacuum for 24 h before use. A mixture of Zn-Salen or Zn-Salen-COF-SDU113 (50 mg) and n-Bu$_4$NBr (600 mg, 1.8 mmol) was placed in a vacuum tube. The reaction system were refilled with $CO_2$, and then 1,2-epoxypropane (25 mmol, 1.75 ml) was added to the tube by a syringe. After being stirred for 48 h at 25 °C, the mixture was then dissolved with ethyl acetate and the insoluble solid was collected by centrifuge and washed by fresh $CH_2Cl_2$ and dried by vacuum for cyclic utilization. After reduced pressure distillation, the filtrate generated a pale yellow oily substance. Further purification of the crude product was carried out through column chromatography. The propylene carbonate and 2,3-epoxybutane were confirmed by gas chromatography (Supplementary Figs. 9 and 10).

## Data availability

The authors declare that all the other data supporting the findings of this study are available within the Article and its Supplementary Information files and from the corresponding author upon request. 10994 optimized metal/Salen-COF structures are provided at https://doi.org/10.6084/m9.figshare.12488519. Source data are provided with this paper.

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

## Acknowledgements

This work was supported by the National Key Research and Development Program of China (No. 2017YFA0204800), the National Science and Technology Major Project of the Ministry of Science and Technology of China (No. 2017ZX05036001), the National Natural Science Foundation of China (No. 21525315 and 21721004), and the Fundamental Research Funds of Shandong University (2019HW016 to L.S., 2019GN023 to D.Z. and 2019GN021 to G.Q.R.).

## Author contributions

L.S. and W.Q.D. planned and designed the project. Q.W.D. and L.S. executed the theoretical screening. L.Y. and D.Z. collected the linker group and built the Salen-COF structures. W.Z. and G.Q.R. executed the syntheses and sample activation. W.Z. Y.M.L. and G.Q.R. performed the catalytic activity evaluation, adsorption characterization, chemical, spectroscopic and electrical characterization. Y.H.Z. contributed to the key comments for upgrading the paper. All authors were involved in the writing of the paper.

## Competing interests

The authors declare no competing interests.
