## [Peer Review File · Nature Communications]

Reviewers' comments:

Reviewer #1 (Remarks to the Author):

In this manuscript, the authors conducted simulation prediction of over 10000 COFs, and identified one COF named Zn-Salen-COF-SDU113 for CO₂ cycloaddition. I appreciate their effort of simulation-guided experimental exploration. However, I am unable to recommend the publication of the work according to the following critical comments:

1. In terms of economic cost of CO₂ fixation, the authors should also take the preparation cost of porous materials/catalysts (COFs in this case) into consideration. As compared with leading porous catalysts for CO₂ conversion, the present work does not provide a breakthrough solution toward CO₂ economic fixation as claimed.
2. The authors highlighted ton-scale production/fixation in the manuscript. It will be promising if the authors could demonstrate large-scale production of their identified COF.
3. To further confirm that their simulation studies are appropriate and useful in guiding the COF design, a few more COFs with lower expected performance to Zn-Salen-COF-SDU113 should be synthesized for comparison.
4. While simulation-guided experiments are appreciated, using the identified COF for CO₂ cycloaddition with epoxide falls into routine: many porous materials have been studied for such type of reactions under ambient conditions. The present COF simply cannot compete with those leading ones.
5. The authors claimed that their COF achieved the first cycloaddition of CO₂ to internal epoxide at ambient conditions. I do not think the authors made fair comparison, since their COF contains coordinated metal ions (other COFs are usually purely organic).

Reviewer #2 (Remarks to the Author):

The manuscript reported a nice work of CO₂ chemistry through the combination of theory and experiments. On the theoretical side, authors screened a large library of salen-COFs and concluded a pore enrichment effect, the pore enrichment could improve catalytic efficiency of salen-based COFs in the carbon dioxide fixation reaction. The authors also synthesized salen-COF and confirmed the extraordinary catalytic efficiency towards cycloaddition of CO₂ to epoxides including internal and terminal under ambient conditions. This work is very interesting in CO₂ chemistry and provides an impressive strategy to design the materials with desired properties aided by computation. Therefore, I would recommend its publication after the following issues are addressed:

- 1) In Figure 1, the labels of y-axis should be provided.
- 2) To make readers clear, the color legend of average densities for CO₂ in Figure 2 and Figure 3a should be given. Also, the labels in y-axis are vague, more clear figures should be provided.
- 3) Some of the number of the figures in the supporting information is not consistent with the manuscript, so it needs to be modified.
- 4) Details for Pawley refinement and diffracted parameters should be shown in Figure S5.

5) The title for Table S3 should be expressed in an appropriate way.

Reviewer #3 (Remarks to the Author):

In the paper by prof Wei-Qiao Deng and co-workers, according to the theoretical investigation of about more than ten thousands of M-salen-COF compounds, the Zn-Salen-COF-SDU113 was chosen, prepared and its catalytic activity investigated in the CO₂ fixation procedure.

The results are new, interesting, and the work is well written. I strongly recommended the publication of the present work. The only comment is rewriting of the abstract part.

Correspondence for reviewer #1:

Reviewer 1: In this manuscript, the authors conducted simulation prediction of over 10000 COFs, and identified one COF named Zn-Salen-COF-SDU113 for CO₂ cycloaddition. I appreciate their effort of simulation-guided experimental exploration. However, I am unable to recommend the publication of the work according to the following critical comments:

1. In terms of economic cost of CO₂ fixation, the authors should also take the preparation cost of porous materials/catalysts (COFs in this case) into consideration. As compared with leading porous catalysts for CO₂ conversion, the present work does not provide a breakthrough solution toward CO₂ economic fixation as claimed.

To: In terms of economic cost of CO₂ fixation, the authors should also take the preparation cost of porous materials/catalysts (COFs in this case) into consideration.

Response: Thank you for your constructive advice. For COF catalysts, the cost is a significant factor limiting their industrial application. Therefore, we analyzed the cost of large-scale synthesis of COF catalysts. Based on the estimation of the synthetic route shown below, the cost of precursor materials for producing a ton of Zn-Salen-COF-SDU113 will be approximately 27000 US dollars. This cost for a recyclable catalyst is affordable for large-scale industrial applications.

To: As compared with leading porous catalysts for CO₂ conversion, the present work does not provide a breakthrough solution toward CO₂ economic fixation as claimed.

Response: Thanks for your comment. We partly disagree with this point. We checked out related literatures with 207 publications and concluded that the present work provides a leading porous catalyst for CO₂ conversion. As shown in Table S3, the obtained Zn-Salen-COF-SDU113 is comparable to the leading porous catalysts for CO₂ cycloaddition to PO. As summarized in Table S3, only one reported porous crystalline material (Cu-MOF 1)

shows comparable catalytic performances (in part catalytic performance parameter) with the identified COF (Zn-Salen-COF-SDU113). In entry 5, the Cu-MOF 1 shows a higher TON value (9600) than Zn-Salen-COF-SDU113 (3068.9), but the reaction yield of Cu-MOF 1 (96 %) is lower than Zn-Salen-COF-SDU113 (98.2 %). Moreover, Zn-Salen-COF-SDU113 achieved the first cycloaddition of CO₂ to internal epoxide at ambient conditions among all COFs and present the highest catalytic performance towards cycloaddition of CO₂ to 2, 3-epoxybutane among all porous materials including MOFs, HCPs, COFs, and CMPs.

The reason we can find this leading catalyst is that the finding of “pore enrichment effect” assisted us to use computational simulation to rapidly screen out the lead from 10994 COF candidates. Therefore, our work provides a potential breakthrough solution toward CO₂ economic by developing a leading porous catalyst. Our description in the abstract of previous version may lead to misunderstanding, and we modified the abstract accordingly.

Table S3. Comparison of cyclic carbonate yields of different porous crystalline materials.

Entry	Catalysts	TBAB (mmol)	Time (h)	Yield (%)	TOF	TON	Ref
1	Zn-Salen-COF-SDU11	1.8 mmol	48	84.2	5.95	1142.7	This work
2	Zn-Salen-COF-SDU113	1.8 mmol	48	97.3	63.35	3040.6	
3	Zn-Salen-COF-SDU113	1.95 mmol	48	98.2	63.94	3068.9	
4	HKUST-1	1.95 mmol	36	65	135.4	4874.4	JACS, 2016, 138, 2142
5	Cu-MOF 1	1.95 mmol	48	96	200 [†]	9600	
6	HKUST-1	1.95 mmol	36	30.0	15.5 [†]	558	Nat. Commun., 2019, 10, 2779
7	CASFZU-1	1.95 mmol	36	98.0	54	1944	

Note: TON results based on per metal active sites (paddlewheel Cu₂ cluster or per Zn active sites).

2. The authors highlighted ton-scale production/fixation in the manuscript. It will be promising if the authors could demonstrate large-scale production of their identified COF.

Response: According to your advice, we have synthesized the identified COF (Zn-Salen-COF-SDU113) on a larger scale and addressed a new approach to enlarge the production of Zn-Salen-COF-SDU113. For the synthesis of 50.3 mg COF, we use the tube sealing method. In the synthesis of 1 g COF, we use the hydrothermal reactor to synthesize it. The latter method is simple and feasible, and has the potential to expand the synthesis further. By increasing the dosage of organic ligands by 20 times, we found that the synthesized Zn-Salen-COF-SDU113 can still keep good crystallinity.

The synthesis method is shown in the METHODS part of the revised manuscript, as follow.

For the synthesis of 1.02 g COF, 1 g TTHEPB were dissolved in a mixture of 32 ml 1,4-dioxane and 8 ml EtOH. Then, 200 μ l EDA was added, the mixed solution quickly turned green, 2 ml HOAc (3 M) is immediately added to the above solution. The solution was further moved into a 100 mL stainless autoclave and heated at 120 $^{\circ}$ C for 3 days. The solid was collected by centrifugation and Soxhlet extraction with dried 1,4-dioxane and acetone. The powder was then dried at 80 $^{\circ}$ C under vacuum for 24 h to give the Salen-COF in 85 % isolated yield.

The characterization results are shown in Figure S5 of the revised supplementary file, as follows.

For the comment “ton-scale production/fixation highlighted in the manuscript”, we are sorry that the abstract of previous version may have misleading contents. We haven’t completed any reaction at such scale. To avoid misleading readers, we have revised the

abstract carefully in the current version of the manuscript.

The revised abstract: Chemical fixation of carbon dioxide (CO₂) may be a pathway to retard the current trend of rapid global warming. However, the current economic cost of chemical fixation remains high because the chemical fixation of CO₂ usually requires high temperature or high pressure. The rational design of an efficient catalyst that works at ambient conditions might substantially reduce the economic cost of fixation. Here, we report the rational design of covalent organic frameworks (COFs) as efficient CO₂ fixation catalysts under ambient conditions based on the finding of “pore enrichment”, which was concluded by a detailed investigation of the 10994 COFs. The best predicted COF, Zn-Salen-COF-SDU113, was synthesized, and its efficient catalytic performance for CO₂ cycloaddition to terminal epoxide was confirmed with a yield of 98.2 % and turnover number (TON) of 3068.9 under ambient conditions, which is comparable to the reported leading catalysts. Moreover, this COF achieved the first cycloaddition of CO₂ to 2, 3-epoxybutane under ambient conditions among all porous materials. This work provides a strategy for designing new porous catalysts in the economic fixation of carbon dioxide.

3. To further confirm that their simulation studies are appropriate and useful in guiding the COF design, a few more COFs with lower expected performance to Zn-Salen-COF-SDU113 should be synthesized for comparison.

Response: According to your suggestion, we have synthesized a new COF (Zn-Salen-COF-SDU11). Its details of structure information and CO₂ cycloaddition performances have been supplied in Table S1 and S3, respectively. The simulated results show that the internal pressure and increment of reaction rate of Zn-Salen-COF-SDU11 are both lower than Zn-Salen-COF-SDU113 under room temperature and atmospheric pressure. The experimental results confirmed that this Zn-Salen-COF-SDU11 shows catalytic activity with a yield of 84.2 % and a turnover number (TON) value of 1142.7, which are much lower than Zn-Salen-COF-SDU113 (Yield: 98.2 %, TON: 3068.9). Both the theoretical and experimental results are in good agreement with each other. The corresponding content has

been added in Page 12 of the revised manuscript and in Figure S1 and S3 of the revised supplementary files.

Simulated results and synthesis conditions for Zn-Salen-COF-SDU11 and Zn-Salen-COF-SDU113

Entry	Simulated Results	Ligands	Solvents &Temp.	Structure
1	P_{in} : 25.3 bar I_k : 29.3	TTHEPB / EDA 0.055 mmol / 0.45 mmol 	6.4 ml THF 1.6 ml EtOH 0.1 ml (3 M) HOAc 120 °C, 3 days	Salen-COF-113 2	P_{in} : 6.0 bar I_k : 6.2	THEPB 0.3 mmol / 0.45 mmol 	3.2 ml 1,4-Dioxane 0.8 ml EtOH 0.2 ml (3 M) HOAc 120 °C, 3 days	Salen-COF-11 
* P_{in} is the internal pressures in the sorption simulations. I_k is the increment of reaction rate under the pore enrichment effect.

Comparison of catalytic performance of two COFs.

Entry	Catalysts	TBAB	Time (h)	Yield (%)	TOF	TON
1	Zn-Salen-COF-SDU11	1.8 mmol	48	84.2	5.95	1142.7
2	Zn-Salen-COF-SDU113	1.8 mmol	48	97.3	15.84	3040.6

Note: TON results based on per metal active sites (paddlewheel Cu_2 cluster or per Zn active sites).

4. While simulation-guided experiments are appreciated, using the identified COF for CO_2 cycloaddition with epoxide falls into routine: many porous materials have been studied for such type of reactions under ambient conditions. The present COF simply cannot compete with those leading ones.

Response: Thanks for your comment. We checked out the related literature with 207 publications and concluded that the present work exhibits a leading porous catalyst for CO₂ conversion. For the reaction of carbon dioxide with PO, as summarized in Table S3, only one reported porous crystalline material (Cu-MOF 1) shows comparable catalytic performances (in part catalytic performance parameter) with the identified COF (Zn-Salen-COF-SDU113). In entry 5, the Cu-MOF 1 shows a higher TON value (9600) than Zn-Salen-COF-SDU113 (3068.9), but the reaction yield of Cu-MOF 1 (98.2 %) is lower than Zn-Salen-COF-SDU113 (96 %). Although not all performance parameters of Zn-Salen-COF-SDU113 are better than the leading porous catalysts, our COF is still comparable with some performance parameters even better than the leading one. For the reaction of carbon dioxide cycloaddition to 2, 3-epoxybutane, Zn-Salen-COF-SDU113 exhibits the best catalytic performances including all porous catalyst. For overall cycloaddition reactions, we believe this Zn-Salen-COF-SDU113 is one of the leading catalysts. We have revised our manuscript to make a clear claim, especially in the abstract part. The corresponding comparisons of cyclic carbonate yields were listed in Figure S3 of the revise supplementary files.

Table S3. Comparison of cyclic carbonate yields of different porous crystalline materials.

Entry	Catalysts	TBAB (mmol)	Time (h)	Yield (%)	TOF	TON	Ref
1	Zn-Salen-COF-SDU11	1.8 mmol	48	84.2	5.95	1142.7	This work
2	Zn-Salen-COF-SDU113	1.8 mmol	48	97.3	63.35	3040.6	
3	Zn-Salen-COF-SDU113	1.95 mmol	48	98.2	63.94	3068.9	
4	HKUST-1	1.95 mmol	36	65	135.4	4874.4	JACS. 2016 , 138, 2142
5	Cu-MOF 1	1.95 mmol	48	96	200 [†]	9600	
6	HKUST-1	1.95 mmol	36	30.0	15.5 [†]	558	Nat. Commun. , 2019 , 10, 2779
7	CASFZU-1	1.95 mmol	36	98.0	54	1944	

Note: TON results based on per metal active sites (paddlewheel Cu₂ cluster or per Zn active sites).

5. The authors claimed that their COF achieved the first cycloaddition of CO₂ to internal epoxide at ambient conditions. I do not think the authors made fair comparison, since their COF contains coordinated metal ions (other COFs are usually purely organic).

Response: Thanks for your comments. We made a detailed literature search and carefully read 207 related literature. We use keywords such as “CO₂ and internal epoxide”, “CO₂ and epoxide and room temperature” and search them in web of science to obtain these 207 publications. To best of our knowledge, the reported catalysts constructed by purely organic ligands and other COFs contains coordinated metal ions do not exhibit CO₂ cycloaddition performances towards internal epoxide at ambient conditions. Moreover, our identified COF achieved the first cycloaddition of CO₂ to internal epoxide at ambient conditions among all COF materials and present the highest catalytic performance towards cycloaddition of CO₂ to 2, 3-epoxybutane among all porous materials including MOFs, HCPs, COFs, and CMPs. The search results related to our reaction are summarized in the following table:

Catalysts	substrate	conditions	Time	Yield	Ref.
PSIL-NTf2		100 °C, 0.8 MPa	8 h	78 %	Catal. Today , 2016 , 265, 56
L-tryptophan		120 °C, 2 MPa	24 h	95 %	Catal. Commun. , 2014 , 44, 6
Co-Salen/DMAP		100 °C, 2 MPa	12 h	95 %	Chem. Commun. , 2004 , 1622
[K ⁺ {PEG}I ⁻]		120 °C, 1 MPa	18 h	79 %	RSC Adv. , 2018 , 8, 30860
Co-MOF-1		100 °C, 3 MPa	8 h	96 %	J. Mater. Chem. A , 2019 , 7, 2884
Cu-MOF		25 °C, 0.1 MPa	24 h	92 %	Chem. Commun. , 2017 , 53, 13371
Zn-Salen-COF-SDU113		25 °C, 0.1 MPa	48 h	74 %	This work

Correspondence for reviewer #2:

Reviewer 2: The manuscript reported a nice work of CO₂ chemistry through the combination of theory and experiments. On the theoretical side, authors screened a large library of salen-COFs and concluded a pore enrichment effect, the pore enrichment could improve catalytic efficiency of salen-based COFs in the carbon dioxide fixation reaction. The authors also synthesized salen-COF and confirmed the extraordinary catalytic efficiency towards cycloaddition of CO₂ to epoxides including internal and terminal under ambient conditions. This work is very interesting in CO₂ chemistry and provides an impressive strategy to design the materials with desired properties aided by computation. Therefore, I would recommend its publication after the following issues are addressed:

1) In Figure 1, the labels of y-axis should be provided.

Response: Thanks for your suggestion. We have provided the labels of the y-axis in Figure 1.

Figure 1. Diagram of 10994 metal/Salen-COF structures and the excess adsorption (mol/g) of CO₂ at 298 K and under 1 bar. The metal elements in the blue background are those used in M-Salen.

2) To make readers clear, the color legend of average densities for CO₂ in Figure 2 and Figure 3a should be given. Also, the labels in y-axis are vague, more clear figures should be provided.

Response: According to your advice, in Figure 2 and Figure 3a, we have added a color legend of average densities. And the labels in y-axis of Figure 3 have been modified.

Figure 2:

Figure 3:

3) Some of the number of the figures in the supporting information is not consistent with the manuscript, so it needs to be modified.

Response: According to your advice, we have carefully revised the number of figures in the manuscript and supporting information.

4) Details for Pawley refinement and diffracted parameters should be shown in Figure S5.

Response: According to your advice, we have provided the details for Pawley refinement and diffracted parameters of Salen-COFs in Figure S5.

5) The title for Table S4 should be expressed in an appropriate way.

Response: Thanks for your advice. We have revised the title for Table S3.

Correspondence for reviewer #3:

Reviewer 3: In the paper by prof Wei-Qiao Deng and co-workers, according to the theoretical investigation of about more than ten thousands of M-salen-COF compounds, the Zn-Salen-COF-SDU113 was chosen, prepared and its catalytic activity investigated in the CO₂ fixation procedure.

The results are new, interesting, and the work is well written. I strongly recommended the publication of the present work. The only comment is rewriting of the abstract part.

Response: Thanks for your advice. We have rewritten the abstract part. The revised abstract part is shown below:

Chemical fixation of carbon dioxide (CO₂) may be a pathway to retard the current trend of rapid global warming. However, the current economic cost of chemical fixation remains high because the chemical fixation of CO₂ usually requires high temperature or high pressure. The rational design of an efficient catalyst that works at ambient conditions might substantially reduce the economic cost of fixation. Here, we report the rational design of covalent organic frameworks (COFs) as efficient CO₂ fixation catalysts under ambient conditions based on the finding of “pore enrichment”, which was concluded by a detailed investigation of the 10994 COFs. The best predicted COF, Zn-Salen-COF-SDU113, was synthesized, and its efficient catalytic performance for CO₂ cycloaddition to terminal epoxide was confirmed with a yield of 98.2 % and turnover number (TON) of 3068.9 under ambient conditions, which is comparable to the reported leading catalysts. Moreover, this COF achieved the first cycloaddition of CO₂ to 2, 3-epoxybutane under ambient conditions among all porous materials. This work provides a strategy for designing new porous catalysts in the economic fixation of carbon dioxide.

REVIEWERS' COMMENTS:

Reviewer #1 (Remarks to the Author):

The manuscript has been substantially revised according to reviewers' comments. A minor revision is recommended before the acceptance.

1. In Figure 4c,d, please make sure that the enlarged images were obtained from the indicated areas. Please also add scale bars to the enlarged images.
2. In the response letter, the authors made comparison with a literature paper (JACS 2016, 138, 2142). It is reasonable to compare their metal-coordinated COF with MOF materials. Since this JACS paper presents an important benchmark for comparison, it should be cited and discussed in the main manuscript.

Reviewer #2 (Remarks to the Author):

MYy questions are well solved, I recommend its publication as it is.

Reviewer #1 (Remarks to the Author):

The manuscript has been substantially revised according to reviewers' comments. A minor revision is recommended before the acceptance.

1. In Figure 4c, d, please make sure that the enlarged images were obtained from the indicated areas. Please also add scale bars to the enlarged images.

Response: Thanks for the suggestion. According to the advice, we have confirmed that the enlarged images were obtained from the indicated areas, and we have added scale bars to the enlarged images.

2. In the response letter, the authors made comparison with a literature paper (JACS 2016, 138, 2142). It is reasonable to compare their metal-coordinated COF with MOF materials. Since this JACS paper presents an important benchmark for comparison, it should be cited and discussed in the main manuscript.

Response: Thanks for the suggestion. According to the advice, we have cited and

discussed this important work in proper places in the revised manuscript.

Reviewer #2 (Remarks to the Author):

My questions are well solved, I recommend its publication as it is.

Response: Thank you very much for your valuable comments.